# A Novel Methodology for Detecting Variations in Cell Surface Antigens Using Cell-Tearing by Optical Tweezers

**DOI:** 10.3390/bios12080656

**Published:** 2022-08-19

**Authors:** Chih-Lang Lin, Shyang-Guang Wang, Meng-Tsung Tien, Chung-Han Chiang, Yi-Chieh Lee, Patrice L. Baldeck, Chow-Shing Shin

**Affiliations:** 1Graduate Institute of Biotechnology and Biomedical Engineering, Central Taiwan University of Science and Technology, Taichung City 40601, Taiwan; 2Department of Automatic Control Engineering, Feng Chia University, Taichung City 407802, Taiwan; 3Department of Medical Laboratory Science and Biotechnology, Central Taiwan University of Science and Technology, Taichung City 40601, Taiwan; 4General Education Center, Feng Chia University, Taichung City 407802, Taiwan; 5ENSL, CNRS, Laboratoire de Chimie UMR 5182, 46 allée d’Italie, 69364 Lyon, France; 6Department of Mechanical Engineering, National Taiwan University, Taipei City 10617, Taiwan

**Keywords:** antigenic variation, cell surface antigens, agglutination test, whole-cell-based analysis, optical tweezers, cell-tearing

## Abstract

The quantitative analysis of cell surface antigens has attracted increasing attention due to the antigenic variation recognition that can facilitate early diagnoses. This paper presents a novel methodology based on the optical “cell-tearing” and the especially proposed “dilution regulations” to detect variations in cell surface antigens. The cell attaches to the corresponding antibody-coated slide surface. Then, the cell-binding firmness between a single cell and the functionalized surface is assayed by optically tearing using gradually reduced laser powers incorporated with serial antibody dilutions. Groups B and B3 of red blood cells (RBCs) were selected as the experiment subject. The results indicate that a higher dilution called for lower power to tear off the cell binding. According to the proposed relative-quantitative analysis theory, antigenic variation can be intuitively estimated by comparing the maximum allowable dilution folds. The estimation result shows good consistency with the finding in the literature. This study suggests a novel methodology for examining the variation in cell surface antigens, expected to be widely capable with potential sensor applications not only in biochemistry and biophysics, but also in the micro-/nano- engineering field.

## 1. Introduction

The analysis of surface antigens on cells has attracted increasing attention due to antigenic variation recognition being crucial for clinical diagnoses. Numerous antigens composed of peptides, proteins, and other molecules are present on cell surfaces. Antigens are the crucial molecular markers of cell functions and lineages [1]. Quantitative evaluation of antigenic variation is critical, especially for hematologic development, immune response, and tumor progression [2,3,4]. For instance, RBCs can be classified by ABO blood group system based on inherited differences and expression in cell surface antigens [5]. The human leukocyte antigens (or CD antigens) expressed on immune cells participate in or perform functions in the immune response [6]. Many cancer cells affect epithelial-mesenchymal transition (EMT) through variations in surface antigens, driving tumor progression [7,8,9].

Flow cytometry (FC), enzyme-linked immunosorbent assays (ELISA), and quantitative polymerase chain reaction (qPCR) have been developed for the analysis of antigenic variations [10,11,12,13,14]. For instance, Cho et al. revealed that the B antigens expression on B305 allele is 35.5% of B101 [10], while the Bx01 is 11.4%. Chen et al. showed that the B antigen expression on B3 cells is 40.92% of B1 [11]. Even though the conventional methods could achieve quantitative measurement, the need for large clinical samples [15,16] limits the practicability. For example, the FC requires more than 5000 to 10,000 copies of the target antigen on one cell to ensure reliable analysis [15], whereas 50,000 cells per well are needed for ELISA to obtain high optical density values and low background noise during detection [17]. Recently, the focus has shifted to the analysis of the individual whole-cell to minimize sample size, suggesting an alternative approach for detecting the variation in cell surface antigens.

The optical tweezers (OT) and the atomic force microscope (AFM) have been widely employed in single-molecule biophysics research [18,19,20,21]. The OT, in contrast to the AFM, is practical for single cell manipulation with precise force measurement [18]. In 1989, direct trapping of a single cell using OT was reported in the pioneering works of Ashkin et al. [22]. Subsequently, the OT is now being used in the investigation of an increasing number of biochemical and biophysical processes [23], including the manipulation of single cells [24,25,26,27,28,29,30,31]. Distinct advantages of using OT include non-contact cell manipulation, pN force accuracy, and amiability in liquid medium environments [26]. The optical force exerted on the microparticle can be measured by the hydrodynamic drag method, which increases linearly with the laser power [27]. Keloth et al. manipulated single cells by the OT using the range from 1 to 40 pN of optical force, with laser powers of 2 to 40 mW [28]. Grexa et al. used a fancy tool with a 3D structure manipulated by OT to measure the single-cell elasticity [29]. This novel application of OT extends the force range available for cell indentations measurements down to the fN regime. The OT has also been widely employed to study the biological characteristics of RBCs [32,33,34,35,36,37,38]. For instance, Yang et al. used the OT to estimate the interaction between RBCs in the coagulation process [34]. Lee et al., Ermolinskiy et al., and Chen et al. sequentially used the OT to investigate the aggregation and adhesion characteristics of RBCs [35,36,37]. Agrawal et al. applied the dual-OT in the assessment of RBC deformability [38]. After decades of developments, convenient commercial OT machines and automatic manipulation systems with microfluidic chips [39,40,41] are becoming available and popular. Many easy-to-use OT products have been promoted. Although the OT has been widely used as a convenient manipulation tool in interdisciplinary fields, employing this technology as a biosensor using a “tearing” operation is still rare.

In this study, we developed a novel methodology based on the cell-tearing operation and the especially proposed “antibody dilution regulation” to estimate the variation in cell surface antigens. The OT is employed to tear off an individual cell from a functionalized slide surface. Finally, through a serial assay, the antigen variation in the cell surface was intuitively estimated by relative-quantitative analysis. Herein, the measurement of singular antibody-antigen interactions is not included because the crucial issue of cell surface antigens for a patient’s clinical diagnosis is the relative expression of a particular antigen, but not the absolute antigen quantity.

## 2. Materials and Methods

### 2.1. Preparations of RBCs and Antibody-Coated Slides

Some distinctive blood groups (e.g., subgroups) or baby’s blood exhibit the phenomenon that the surface antigens have a weak expression on the RBC surface [42]. For example, A2 antigens approximately express 30% of A1 on the cell surface, and A1 in newborns is 31% of A1 in adults [43]. In this study, the most common B subgroup in the Asian population [11], i.e., the B3, is selected as the subject for the comparison with B (common group) to detect the antigenic variation. Standardized RBCs were provided by Formosa Biomedical Technology Corp. (Taipei, Taiwan) and diluted 800-fold with PBS (Phosphate buffered saline) before the addition of 0.1 g/mL BSA (Bovine serum albumin) to prevent the RBCs from adhering together and block non-specific interactions with antibodies or slides. One drop (~0.02 mL) of the RBCs solution was incubated on the antibody-coated slides for 20 min at room temperature just prior the test.

The antibody-coated slides were prepared by protein adhesion on surfaces coated with poly-L-lysine [44,45], an efficient method to prepare antibody microarrays [46]. Cover slides (Paul Marienfeld GmbH & Co. KG, Lauda-Königshofen, Germany) were cleaned with acidic alcohol (1% HCl in 70% ethanol), rinsed thoroughly in ultra-pure H_2_O, incubated at room temperature in a 1:10 poly-L-lysine solution (#P8920, Sigma-Aldrich, St. Louis, MO, USA) for 5 min, and then dried in a 60 °C oven for 1 h. Solutions of anti-A and anti-B monoclonal antibodies (1 mg/mL) were provided by Thermo Fisher Scientific Inc. (Waltham, MA, USA) and diluted with PBS solutions (#P4417, Sigma-Aldrich, St. Louis, MO, USA) with serial dilution folds. Poly-L-lysine coated slides were incubated in antibody solutions at room temperature for 1 h, then for 5 min in 0.05 g/mL BSA solution to block non-specificity, and stored at 4 °C.

### 2.2. Optical Tweezers for Tearing off an RBC

The schematic of the OT system based on an inverted microscope platform (Olympus IX51) is shown in Figure 1. A continuous-wave Nd-YAG laser (model #ISF064-1000P, Onset Electro-Optics Co., Ltd., New Taipei, Taiwan) at λ = 1064 nm focalized by a high numerical aperture (NA = 1.3) microscope objective /oil (UPLFLN100XO2, Olympus, Tokyo, Japan) provides the trapping beam with maximum available powers of 250 mW. The RBC solutions were confined to an isolated sample chamber comprising two cover slides (170 μm thickness), and a double-faced tape (120 μm thickness) to eliminate flow disturbance. The chamber can be moved with an XYZ-axis nanopositioner (NanoCube^®^P611, Physik Instrument (PI), Karlsruhe, Germany), while the laser is focused at a fixed position in the chamber. The dragging direction and the distance (1 μm) from the slide to the trapping spot center are consistent. The dragging speed is kept low (5 μm/s) for the static test during the cell-tearing experiment, so the solution viscosity can be ignored.

The basic principle of OT for manipulating micron-sized dielectric objects have been described previously [47,48,49]. A laser beam is focused by a high NA of a microscope objective to a spot (~1 μm) in a transparent micro-object, generating an optical trapping force. The force (*F*) can be expressed as *F* = *QnP* [50], where *P* is the laser power, *n* is the relative refractive index, *Q* is a dimensionless parameter related with object dimension, NA, wavelength, polarization, beam profile, and spot size. Essentially, 1 mW of laser power approximately generates 1 pN force for a 1 μm diameter sphere [50]. In this study, the experiment subject, RBC, is 7.5~8.5 μm in diameter, with a width of 1.7~2.2 μm in the ring and 0.5~1 μm at the center [51]. The optical trapping spot exerts at a constant volume at the ring edge of an attached RBC during cell-tearing assays. And, the *n*, the NA, the wavelength, the polarization, the beam profile, and the spot size are consistent in the experiment. Hence, the parameter *Q* is consistent during the experiment, meaning the optical force is linearly proportional to the laser power. That is, the laser power can replace the optical force to evaluate the binding firmness between an RBC and antibody-coated surface, promising a simplified approach.

### 2.3. Verification of Antibody-Antigen Interactions

First, the negative control experiment, i.e., the case of non-specific antibody-antigen interaction was performed. Figure 2a schematically illustrates a non-attached RBC trapped by the OT, showing the RBC is vertically aligned by the optical torque (Figure 2b). Then, the suspended RBC can be freely dragged in solution (Figure 2c) with the threshold power, 4 mW. Figure 2d shows the sequent films of the dragging manipulation. The RBC is dragged by optical tweezers freely in solution. In contrary, when the RBC antigens specifically interact with the associated antibody-coated surface, the RBC attaches to the slide (Figure 3a). Figure 3b represents the RBC stretched by the OT, but it cannot be torn off even using the maximum power (250 mW), meaning the binding firmness is strong enough to oppose the pull of tearing.

The fundamental qualitative test of the specific antibody-antigen interaction was verified using common blood groups A, B, and O. A strict criterion was used for each cell-tearing assay in this study. A complete assay includes five cells, and the trial was repeated five times for each cell, meaning 25 consecutive trials to confirm the binding firmness. Then, the assay was marked as “○”. In contrast, once a cell was torn off in any one of the 25 trials, the assay was marked as “×”. The result shown in Table 1 indicates that the qualitative test using the cell-tearing method is consistent with the standard criterion of specific antibody–antigen interaction.

### 2.4. Antibody Dilution Regulation

The binding between a cell and the antibody-coated surface is caused by antibody–antigen affinity which is related to the antibody density (on the functionalized glass surface) and the antigen density (on the cell surface). Thus, the “antibodies dilution regulation” is especially proposed with associating to the “tearing operation” in this study to examine the binding firmness of the cell thoroughly. The glass slides coated with a serially diluted fold of antibody solutions were prepared. The dilution folds increase by an exponent of 2 but will be adjusted according to the test requests. For example, the fold increases by 512 after 2048-fold and 256 after 6656-fold for more precise tearing assays.

## 3. Results

### Preliminary Assay of Cell-Binding

For preliminary evaluation of the cell-binding, the cell-tearing assays were performed using a constant 250 mW of laser power with gradually increased dilution folds. The results listed in Table 2 show the binding of group B cells can be confirmed until the fold of antibody dilution increases to 4608, while the maximum dilution for B3 is 512. The definite difference identifies the B3 and B distinctly. However, the assessment of quantitative variations in cell surface antigens is still unavailable. More precise detection is needed.

Next, the precise examination of the cell-bonding firmness was accomplished by the alternate variate of the cell-tearing power and the antibody dilution fold. That is, the laser power began a successive reduction during an increased dilution. For instance, the group B cells were torn off when the dilution fold was 5120 at 250 mW in the preliminary assays. At this point, the applied powers successively reduced until the cell could not be torn off, which is marked as the power for confirming the binding. Repeat the procedure by alternately increasing the dilution fold and reducing the laser power until the bond was not firm enough anymore to oppose the pull of tearing with the threshold power (4 mW). Then, the assay was terminated. In this experiment, the laser power was gradually reduced by 5 mW to 10 mW, then by 1 mW to the end.

The results listed in Table 3 indicate that the binding of group B cells can be confirmed until 7168-fold dilution, while the maximum dilution for group B3 cells is 2560. As a result, a more distinct difference between B and B3, and the relationships between dilution folds and laser powers were concluded. It is worth noting that the entire experiment was carried out three times with identical results which indicate a very small variability, validating reproducibility.

For comparison, the conventional agglutination method was carried out using the same dilution regulations. The results listed in Table 4 show that group B can be recognized until 256-fold dilution, while group B3 is 128-fold. This concludes that the sensitivity of the cell-tearing method is much higher than the conventional agglutination method. Notably, even if the agglutination method has incorporated the proposed dilution regulation, it only provided a qualitative distinction of group B and B3, but not a quantitative examination of variations in cell surface antigens.

## 4. Discussions

### 4.1. Binding Phenomenon between a Cell and the Functionalized Surface

As previously mentioned, the binding between a cell and the slide surface is related to the cell surface antigen and the antibody on the functionalized slide. Initially, the number of antibodies on the slide was supposed to be saturated and sufficient to interact with antigens on the cell-contacting portion. At this point, the amount of antibody–antigen interactions is consistent. The binding of a cell to the antibody-coated surface is firm enough to oppose the pull of optically tearing even using the highest laser power (250 mW). In the sequent tearing assays, as the antibodies concentration decrease by diluting, the amount of the antibody-antigen interactions is gradually decreasing until the cell can be torn off from the slide surface, i.e., the antibody dilution fold reaches a critical value. Thus, the cell binding is not firm enough to oppose the tearing.

Subsequently, the laser power starts successively reducing, and the antibody concentration continually decreases by dilution. The purpose is to find the relationship between the dilution fold and the laser power by repeating the procedures of alternately increasing the dilution fold and reducing the laser power. Eventually, the binding is not firm enough to oppose the pull of optically tearing even with the threshold power, and then, the assay is terminated. At this point, the antibody dilution fold reached the maximum allowable value for firming the cell bonding.

### 4.2. Estimation of the Variation in Cell Surface Antigens

In this study, the estimation theory of the variation in cell surface antigens is based on a relative-quantitative analysis. The critical dilution folds (e.g., 5120, 5632, 6144, …, etc.) and their associated laser powers (e.g., 145, 25, 10, …, etc.) in Table 3 are used to plot the chart in Figure 4, showing the antibody dilution fold (*D*) is inversely proportional to the laser power (*P*), i.e., *D*~1/*P*. The R-squares of group B (Figure 4a) and group B3 (Figure 4b) are 0.98 and 0.99, respectively, validating the theory.

As mentioned early, the binding between an RBC and the functionalized surface is related to the antibody-antigen affinity. It is reasonable to assume that the integration probability (*R*) of the antigen-antibody is proportional to the densities of the antibody (*d*_1_) and the antigen (*d*_2_), i.e., *R*~*d*_1_*d*_2_, where *d*_1_ = *m*/*A*_1_, *d*_2_ = *n*/*A*_2_; *A*_1_ and *A*_2_ are the areas of the antibody-coated surface and the cell surface, respectively; *m* and *n* are the amounts of antibodies and antigens, respectively. Thus, *R_B3_/R_B_* is equal to $\frac{m_{B3}}{A_{1}}\frac{n_{B3}}{A_{2}}/\frac{m_{B}}{A_{1}}\frac{n_{B}}{A_{2}}$, i.e.,
(1)$$\frac{R_{B3}}{R_{B}}=\frac{m_{B3}n_{B3}}{m_{B}n_{B}},$$

The force (*F*) for tearing off a cell from the functionalized surface should be proportional to the binding probability (*R*). As aforementioned, the force (*F*) can be expressed as the laser power (*P*), and the antibody amount (*m*) is inversely proportional to the dilution fold (*D*). Therefore, the Equation (1) can be rewritten as $\frac{P_{B3}}{P_{B}}=\frac{D_{B}n_{B3}}{D_{B3}n_{B}}$, or
(2)$$\frac{n_{B3}}{n_{B}}=\frac{D_{B3}P_{B3}}{D_{B}P_{B}},$$

When the dilution fold reaches to the maximum at which the cell cannot be torn off with the lowest laser power, e.g., 5 mW (Table 3), the variation in cell surface antigens can be intuitively estimated by comparing the maximum allowable dilutions, that is $\frac{n_{B3}}{n_{B}}=\frac{D_{B3,\max}}{D_{B,\max}}$. With the above relative-quantitative analyses, the maximum allowable antibody dilution in the group B assays is 7168 (*D_B,max_*), while that in group B3 is 2560 (D*_B3,max_*). Thus, the antigens on the group *B3* are 35.7% (*D_B3_,_max_/D_B,max_* = 2560/7168) of that on the group *B*. The result is consistent with the literature using conventional biological methods (Table 5).

The advantage of this approach is that the estimate can be intuitively implemented without complicated calculations of optical force since the strength measurement of a singular antibody-antigen interaction is unnecessary due to the relative antigen expression being crucial rather than the absolute antigen quantity. The operation of cell-tearing is like simply tearing a tape in daily life. Nevertheless, with this simple operation, the quantitative examination of the antigenic variation in the cell surface can be accomplished. It is worth recalling that a minor increase in dilution fold, i.e., more tearing assays, is expected to lead to a higher resolution. In this case, the previously mentioned easy-to-operate automated manipulation systems of optical tweezers may meet the needs. Then, a more accurate assessment of the antigen variation in the cell surface can be achieved.

## 5. Conclusions

This study proposes a new application of optical tweezers as a biosensor. A simple and intuitive methodology using the optical cell-tearing operation has been developed to estimate the variation in cell surface antigens, which is crucial for clinical diagnoses. Since an absolute antigen quantity is unnecessary in the relative-quantitative analysis, the proposed concept can be implemented without complicated optical force calculations and labored biological processing. In addition, one drop of blood from the fingertip is more than enough to obtain a satisfactory assessment. The experiment result is consistent with literature findings that used conventional biological methods. A higher resolution could be achieved with more minute dilution fold increases as needed. This suggests an alternative approach based on the whole-cell operation for antigenic variation analyses. The achievement of this approach is expected to be widely capable with potential sensor applications not only in biochemistry and biophysics, but also in the micro-/nano- engineering field.

## Figures and Tables

**Figure 1 biosensors-12-00656-f001:**
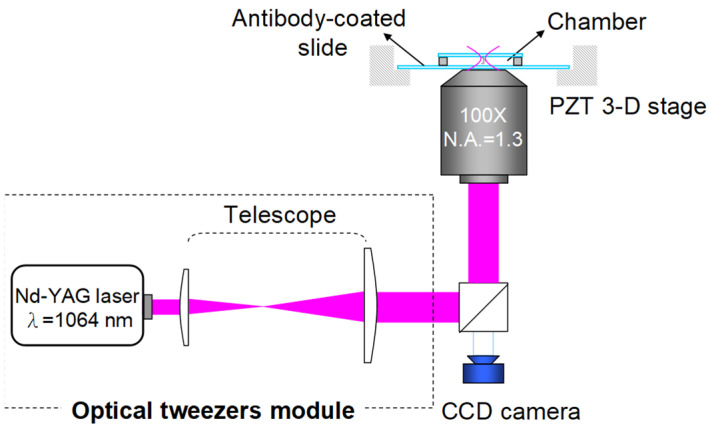
The schematic figures of the OT and the sample chamber.

**Figure 2 biosensors-12-00656-f002:**
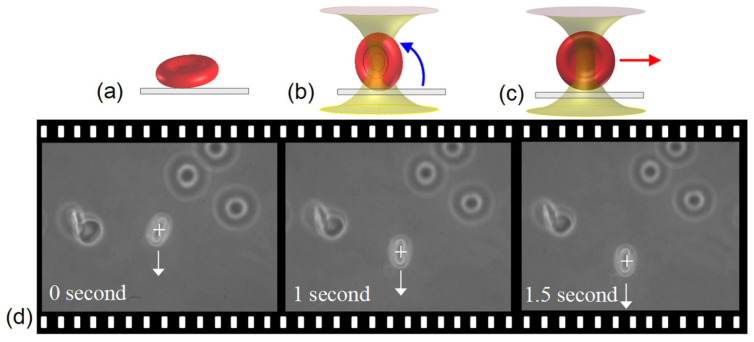
(**a**–**c**) schematic illustration and (**d**) the manipulation films (Appendix A) of an RBC.

**Figure 3 biosensors-12-00656-f003:**
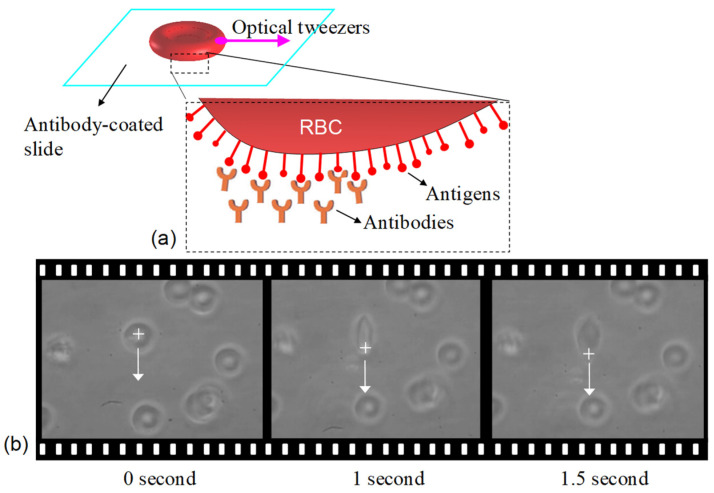
(**a**) Schematic illustration (not in scale) detailing RBC attachment to the antibody-coated surface. (**b**) RBC is torn off by OT but still attaches to the antibody-coated surface (Appendix A).

**Figure 4 biosensors-12-00656-f004:**
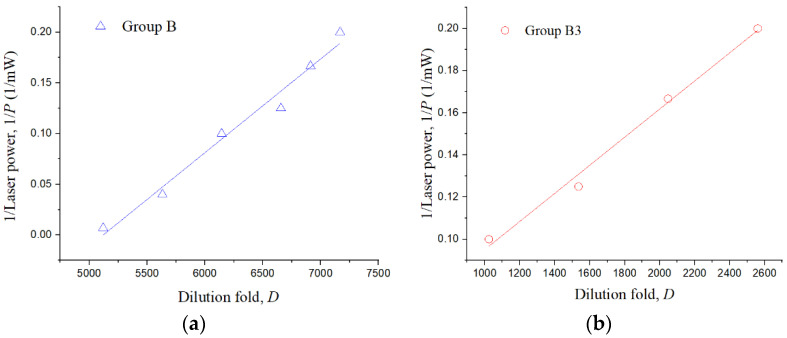
The relationship between fold dilutions (*D*) and laser powers (*P*). (**a**,**b**) are group B and group B3 cells, respectively.

**Table 1 biosensors-12-00656-t001:** Qualitative test of the standard specific antibody-antigen interaction using cell-tearing assay *.

RBC Groups	Anti-A Surface	Anti-B Surface
A	○	×
B	×	○
O	×	×

* ○: bonding confirmed; ×: RBC torn off.

**Table 2 biosensors-12-00656-t002:** Cell-tearing assays with a serial antibody dilution using maximum laser power (250 mW).

Antibody Dilution Fold	Group B Cells	Group B3 Cells
1, 2, 4,…, 256	○	○
**512**	○	○
1024	○	×
2048	○	(Terminated)
2560	○	
3072	○	
3584	○	
4096	○	
**4608**	○	
5120	×	
	(Terminated)	

○: bonding confirmed; ×: RBC torn off.

**Table 3 biosensors-12-00656-t003:** Cell-tearing assays with a serial antibody dilution using gradually decreased powers.

Group B Cells		Group B3 Cells	
Antibody Dilution Fold	Power for Confirming Binding (mW)	Antibody Dilution Fold	Power for Confirming Binding (mW)
5120	145	1024	10
5632	25	1536	8
6144	10	2048	6
6656	8	2560	5
6912	6		(Terminated)
7168	5		
	(Terminated)		

**Table 4 biosensors-12-00656-t004:** Conventional agglutination assays incorporated with a serial antibody dilution. The mark “+” and “−” mean agglutinative and non-agglutinative, respectively.

Antibody Dilution Fold	Group B Cells in Anti-B Antibody Solution	Group B3 Cells inAnti-B Antibody Solution
1	+	+
2	+	+
4	+	+
6	+	+
8	+	+
16	+	+
32	+	+
64	+	+
**128**	+	+
**256**	+	−
512	−	

**Table 5 biosensors-12-00656-t005:** Data of relative antigens expression on RBC surface in the works of literature.

Detection Method	Relative, %	Literature
Flow cytometric analysis	35.5	Cho et al. [10]
RT-qPCR	40.9	Chen et al. [11]
Antibody dilution and cell-tearing	35.7	This study

## Data Availability

Not applicable.

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
