# Peer review of "A Novel Methodology for Detecting Variations in Cell Surface Antigens Using Cell-Tearing by Optical Tweezers"

_biosensors, 2022, doi:10.3390/bios12080656_

Round 1
Reviewer 1 Report
The manuscript introduces a new method for the evaluation of the relative antigen density on the surface of red blood cells. The authors use optical tweezers to directly trap and tear off RBCs from surfaces previously treated with various dilutions of RBC-specific antibody. They provide an intuitive model to calculate the relative densities from the dilution fold and the threshold laser power required for tearing off the cells.
The Introduction cites papers from the literature of optical trap-assisted RBC investigations (although the mentioned Ashkin 1987 paper, ref. 22, does not trap RBCs), which show that the force that builds up during the aggregation of two RBCs is in the range of a few (<10) pNs. However, Introduction does not mention key optical trapping papers that deal with RBC deformation, which, I believe, would be an important aspect of the mehodology of the presented work. From these papers (Bronkhorts 1995, Rao 2009, Steffen 2011, Zhu 2019) it turns out that only a few tens of mW laser powers with similar focusing objectives and wavelengths which was used in the mansucript, can already deform (bend, stretch) the cells considerably. The authors used up to 250mW power, therefore their system is capable of deforming the cells by far.
I have three main concers about the methodology:
1) For tearing off the RBC from the substrate, the authors first trap the cells then pull them sideways when it either detaches from the surface or not. Now, considering the high laser powers used, it is safe to assume that cells, that are attached to a surface by a portion of their membranes, deform when they are pulled sideways with the trap.
Which may easily have two consequencies: one, the cell-substrate contact area can change during the tearing off, which naturally modifies its threshold power. Second, the Q factor mentioned in the method (line 125) and in the model (line 240), does change during the pulling, therefore trapping force is not linearly changes with the power any more. This deformation and its consequencies must be investigated and addressed in the manuscript.
2) The authors assume as the basis of their model, that the integation probability (line 233) (which is called binding probability in line 240) is proportional to the product of the antigen and antibody surface densities: R ~ d1 * d2 (line 234). It might be valid for close-to-equal concentrations. However, when they differ substantially, the limiting factor for binding will be the smaller density. For instance, doubling the already higher density will not double probability because on the other surface there's no partner for the interaction. In the manuscript, the antibody concentration on the substrate spans more than 3 orders of magnitude with the dilution method, so the case of very different densities must be handled in the paper.
3) The model leads to a sympathetically simple solution where at a given laser power the threshold dilutions of the two kinds of antibodies for the tearing off povide the relative antigen conentration on the red blood cells. The authors stopped at 5mW as the minimum laser power. This choice seems arbitrary to me, since they could have investigated the dilution fold at 4mW or 3mW also. The importance of the proper laser power is evident from Table 3: from the threshold dilution folds at 6mW or 8mW, the relative antigen concentrations become 29.6% and 23% respectively, much lower that the one found 35.7% for 5mW (line 249). The choice of the laser power must be justified better in the manuscript.
Author Response
Thank you very much. Your professional comments help to strengthen this manuscript. The responses are enclosed in the attachment. Thanks a lot for your time.

Reviewer 2 Report
This manuscript can be accepted to publish in Biosenairs.
Author Response
Thank you very much for your appreciation of this study. Your professional comments help to strengthen this manuscript.

Reviewer 3 Report
1. Need to include literature review on optical tweezers force scope of single cell in the introduction. There are many work on cancer cells and RBC.
2. Page 3 line 125. It is not totally correct to say Q is constant, as for different trapping position on the cell the trapping volume would be different and the trapping force would be different. If the RBC is off from its balanced trapping location the force would be different. Please discuss it.
3. Please theatrically estimate trapping force for different trapping power with different offset positions.
4. It is suggested to experimentally measure the trapping force, using the momentum conservation method or using drag force (which is quite easy to do) method.
Author Response
Thank you very much. Your professional comments help to strengthen this manuscript. The responses according to your comments and suggestions are enclosed in the attachment. Thank you in advance for your time.

Reviewer 4 Report
Lin and co-workers developed a method to quantify the cell surface antigen by using optical tweezer technique. Red blood cells on a surface functionalized with different coverages of antibody were pulled by the optical tweezer, and the rupture of the cells from the surface was recorded for antigen quantification. Although the data can adequately support the conclusion, the paper still need a minor revision before it can be published. Please see my comments below:
1. The paper has too many tables in it. I had a hard time reading the tables and trying to extract the useful information. Can the authors use figures instead of the long tables (especially Table 3) to present their result and make it more clearly?
2. Page 2 line 89. The antibodies were immobilized on the surface using poly-L-lysine. The interaction between antibody and poly-L-lysine is non-covalent bond or physical absorption, which is weak and easy to break. How can the authors know that they are breaking the antibody-antigen bond rather than the antibody-poly-L-lysine bond?
3. Page 2 line 97. The antibody was incubated for 1 hour. Did the authors measure the antibody coverage after the incubation? When using high concentrations, it is possible that the surface is saturated in a short time.
4. Page 3 line 121. Does the antigen distribute evenly in the ring region and in the center? Does the orientation of RBC affect the rupture force?
Author Response

(The authors gave the same response as above.)

Round 2
Reviewer 3 Report
N/A
Reviewer 4 Report
All my questions are clearly addressed, I recommend publication